# Bat-G net: Bat-inspired High-Resolution 3D Image Reconstruction using Ultrasonic Echoes

**Gunpil Hwang**[*]**, Seohyeon Kim,**[*] **and Hyeon-Min Bae**
School of Electrical Engineering
Korea Advanced Institute of Science and Technology
Daejeon, South Korea
{gphwang, dddokman, hmbae}@kaist.ac.kr

## Abstract

In this paper, a bat-inspired high-resolution ultrasound 3D imaging system is presented. Live bats demonstrate that the properly used ultrasound can be used to perceive 3D space. With this in mind, a neural network referred to as a Bat-G network is implemented to reconstruct the 3D representation of target objects from the hyperbolic FM (HFM) chirped ultrasonic echoes. The Bat-G network consists of an encoder emulating a bat's central auditory pathway, and a 3D graphical visualization decoder. For the acquisition of the ultrasound data, a custom-made Bat-I sensor module is used. The Bat-G network shows the uniform 3D reconstruction results and achieves precision, recall, and F1-score of 0.896, 0.899, and 0.895, respectively. The experimental results demonstrate the implementation feasibility of a high-resolution non-optical sound-based imaging system being used by live bats. The project web page (https://sites.google.com/view/batgnet) contains additional content summarizing our research.

## 1 Introduction

Recent improvements in sensor and information processing technologies have made significant contributions to the progress of numerous unmanned systems (UMS) such as a drone, an autonomous vehicle, and a robot. In order for UMS to reach full autonomous level that does not require any human intervention, the collected data from sensors in UMS should suffice to manage the entire environmental scenarios. Therefore, UMS commonly employs a combination of sensors including RGB-D cameras, RADARs, LIDARs, and ultrasonic sensors that are complementary to each other. Both RGB-D camera and LIDAR provide abundant high-resolution visual information, however, the visibility and accuracy can be severely compromised depending on environmental/weather conditions as shown in Fig. 1(a). On the contrary, RADAR and conventional ultrasonic sensors, measuring the time-of-flight of the reflected signal, are relatively less sensitive to operating circumstances but merely provide low-resolution ranging information [1, 2]. Consequently, a clear need exists for an imaging sensor that can precisely visualize 3D space irrespective of environmental conditions.

In this paper, a high-resolution ultrasound 3D imaging system emulating the echolocation mechanism of a live bat is presented. Among many outstanding features of a bat's sensory system that enables accurate 3D perception, three following key points are essentials: (1) Bats localize obstacles and discriminate prey by analyzing the echoes of emitted ultrasound pulses, which is called echolocation [3–5]. The emitted ultrasound signal from a bat is frequency-chirped over a wide frequency range, which plays a critical role in recognizing the shape of an object from the echo spectra [6]. Hence, the proposed system employs a frequency-chirped broadcasting ultrasound signal in the range of 20-120 kHz as shown in Fig. 1(b). (2) Echolocation is an inverse problem where the spatial information

---

[*]Equal contribution

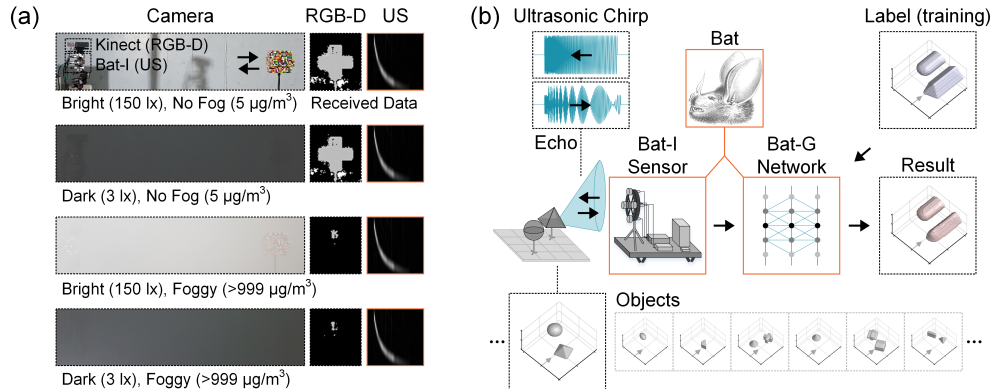

Figure 1: (a) Measurement results of a camera, an RGB-D camera, and an ultrasound (US) sensor in the different brightness/fog level. (b) Overview of a bat-inspired high-resolution ultrasound 3D imaging system.

of a target is extracted from reflected/scattered echoes. In general, solving an inverse problem is an extremely laborious and time-consuming task since the problem is often ill-posed and requires several iterations [7]. However, live bats with a real neural network recognize the surroundings in real time. From these, we can infer the fact that the echolocation problem can be solved efficiently with the help of the artificial neural network. Therefore, we designed a feed-forward neural network to inversely reconstruct a 3D image from the collected ultrasound data, referred to as a bat-inspired graphical visualization (Bat-G) network. (3) The sensory-to-image conversion of bats involves the neural interactions between the nuclei on the central auditory pathway (through *the brainstem* and *the midbrain*) and *the auditory cortex (AC)*. From the sensory input, it is believed that the auditory nuclei extract temporal and spectral features needed for the echolocation and then pass them to AC through monaural, binaural, ipsilateral, and/or contralateral connections. The architecture of Bat-G net is heavily inspired by the neuroanatomical auditory pathway of bats.

## 2 Related Work

In the past decades, airborne ultrasonic sensors have been widely used for range detection. These sensors emit a single frequency ultrasonic signal and calculate the distance to the object in a 2D horizontal plane by measuring the time-of-flight (TOF) of echoes reflected from an object. Recently, there have been attempts to localize/classify a target object and/or reconstruct the shape of an object as shown in table 1. A series of 3D localization strategies have been explored, which includes the calculation of TOF difference between two pairs of microphones [8] and the reception of signals from a designated direction in 3D space using a beamforming (BF) technique [9]. In [10], a biomimetic sonar system performing spectrum-based 3D localization is proposed. Another line of research is the classification of target objects by combining different classification parameters such as the

Table 1: Summary of Related Works

|  | [8] | [9] | [10] | [11] | [12] | [13] | [14] | [15] | [16] | **This Work** |
|---|---|---|---|---|---|---|---|---|---|---|
| 3D localization | ✓ | ✓ | ✓ | ✓ | ✓ | × | ✓ | ✓ | ✓ | ✓ |
| Classification | × | × | × | ✓ | ✓ | ✓ | ✓ | ✓ | ✓ | ✓ |
| Reconstruction | × | × | × | × | × | × | ✓ | ✓ | ✓ | ✓ |
| TX / RX | 1/4 | 1/32 | 1/2 | 3/3 | 4/4 | 1/1 | 1/400 | 1/64 | 5/3 | **1/4** |
| Measurement | ✓ | ✓ | ✓ | ✓ | ✓ | ✓ | ✓ | ✓ | × | ✓ |
| Method[1] | T | BF | SC, BM | T, AC | T, PCA | NN, BM | SA, BF | NN, HG | T, CS | **T, NN, BM** |

NOTE: [1] T: Time difference of arrival, BF: Beamforming, SC: Spectral Cues, BM: Biomimetics, AC: Angle Change, PCA: Principal-Component-Analysis, NN: Neural Network, SA: Synthetic Aperture, HG: Holography, CS: Compressive Sensing

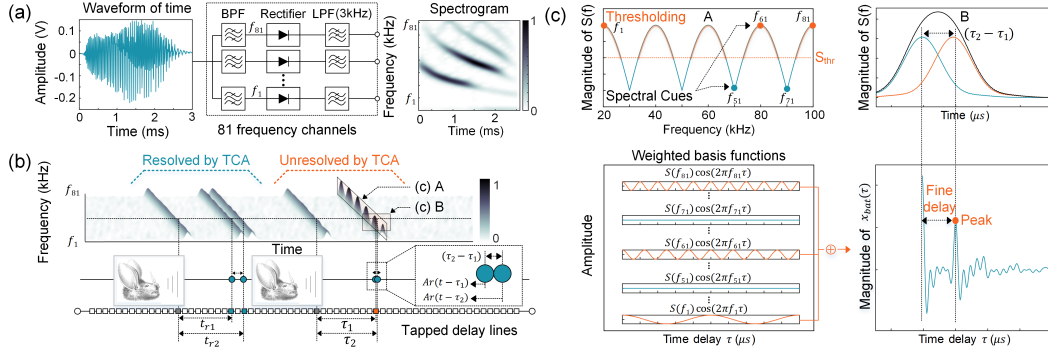

Figure 2: (a) Operational block diagram of a bat's cochlear block. (b) Illustration of the operation of a bat's temporal cue analysis (TCA) block. (c) Fine delay determination mechanism of a bat's spectral cue analysis (SCA) block.

angles/distances between the 3D sensor array and an object [11] or by utilizing 16 TOF vectors (4 TXs and 4 RXs) processed by means of the principal component analysis (PCA) [12]. However, such techniques relying on a lookup table for the classification, distinguish only simple objects such as plane, corner, and edge. On the other hand, [13] has attempted to categorize cube and tetrahedron by analyzing the spectrum of echoes with the help of a neural network (NN). However, the approach has yet to demonstrate the real potential of NN methods due to limited datasets alongside the rudimentary NN structure. Besides 3D localization and classification of target objects, many efforts have been made to solve the ill-posed inverse problem for the reconstruction of the 3D shape of an object from the received echoes. Such attempts adopted either the BF [14] or holography [15] techniques with a large number of TRX array. Compressive sensing (CS) technique, a subset of the inverse problem approach, has also been tried in a simulation domain with a few cuboids considering the sparse property of the scenes [16]. However, such inverse problem approaches require tremendous computation power and time to process the incoming data from such large array. In this paper, a feed-forward Bat-G network is proposed to solve the ill-posed 3D ultrasonic inverse problem. The proposed network reconstructs 3D representation of diverse objects from measured 4-channel ultrasonic signals.

## 3    Preliminaries

In order to understand the 3D spatial perception mechanism of a bat-inspired imaging (Bat-I) sensor, it is essential to understand the structure of a bat's auditory system that comprising three principal components including the cochlear and the temporal/spectral cue analysis block.

### 3.1    Cochlear Block

The position-dependent frequency selectivity of the basilar membrane in the bat's cochlea can be modeled by sharply tuned band-pass filters (BPFs) as described in Fig. 2(a). These filters are typically modeled by 81 parallel constant-bandwidth, 10th-order Butterworth IIR filters whose center frequencies ($f_c$) are hyperbolic in the range of 20-100 kHz. The transmission process linking the excitation of hair cells to the primary auditory neurons through synapse is modeled by half-wave rectification followed by low-pass filtering (LPF) at the output of each 81 BPF [17, 18]. As a result, the emitted/received sound signal is decomposed into 81 band-pass filtered signals and then subsequent rectifier and LPF extract the amplitude (or power) of the signals. Consequently, this process produces the time-frequency representation of the acoustic time-domain signal, which is analogous to the spectrogram.

### 3.2    Temporal/Spectral Cue Analysis (TCA/SCA)

The TCA block measures the elapsed time between the emitted sound signal and its echoes over each repetitive emission time. Delay-tuned neurons operate as tapped delay lines in each frequency channel as described in Fig. 2(b). The emitted and echo signal travel along these delay lines sequentially. Coincidence detection neurons in multiple channels detect the coexistence of the emitted signal and

the echo signal in each tapped delay line. The activated position of the tapped delay lines determines the delay of the echo. When the number of activated channels exceeds the threshold, the location of the target is declared [17].

Fine delay, caused by overlapping echoes reflected from two nearby glints, is unresolvable through direct delay measurements by the TCA block [18]. These fine delays are resolved by the SCA block analyzing the spectral cues such as notch and null. Assuming only two glints exist for simplicity and the echoes $r(t)$ from two glints are reflected back with the same magnitude $A$ but with different delays $\tau_1$ and $\tau_2$, then the received signal $s(t)$ is given by

$$s(t) = Ar(t - \tau_1) + Ar(t - \tau_2), \tag{1}$$

where $t$ denotes time. The frequency spectrum $S(f)$ of $s(t)$ can be written by

$$S(f) = A \cdot R(f)e^{-j2\pi f \tau_1}[1 + e^{-j2\pi f(\tau_2 - \tau_1)}], \tag{2}$$

where $f$ and $R(f)$ are the frequency and the frequency spectrum of the individual echo $r(t)$, respectively. Bats transform the spectral information into a time-delay domain, as shown in Fig. 2(c), by summing up the $|S(f_k)|$-weighted basis only when the magnitude of the frequency spectrum $|S(f_k)|$ of $k$-th frequency channel exceeds the threshold $S_{thr}$ [19, 17, 20–22], namely

$$x_{bat}(\tau) = \sum_{k=1}^{N} |S(f_k)| \cdot \cos(2\pi f_k \tau) \quad if \quad |S(f_k)| > S_{thr}, \tag{3}$$

where $x_{bat}(\tau)$ and $f_k$ denote the time-delay representation of the bats and the center frequency of $k$-th channel, respectively, as shown in Fig. 2(c) [17]. The fine delay is eventually determined by finding the location of the peaks in the time-delay representation. Furthermore, a target can be considered as an object containing several glints and reflecting surfaces [23–26]. Echoes reflected from these glints contribute to the spectral cues of an echo [27]. That is, the shape of a target is expressed with a unique spectral fingerprint. Bats are known to use these spectral signatures to recognize the shape of a target [18, 28, 29]. Consequently, the sophisticated pattern recognition of the spectral cue is central to the spatial perception mechanism.

## 4 Data Acquisition

Bat-inspired imaging (Bat-I) sensor (see Fig. 1(b)) emits broadband FM signals and records echoes reflected from the target object. The recorded data transformed into spectrogram are fed into the Bat-G network for training and the network eventually infers the object's 3D representation. In order to train the network, we have adopted a supervised learning algorithm and created 4-channel ultrasound echo dataset, ECHO-4CH (49 k data for training and 2.6 k data for evaluation). Each echo data consists of eight spectrograms ($256^2$ grayscale image) and one 3D ground-truth label ($64^3$ voxels).

### 4.1 Data (4-channel Ultrasound Echo)

**System Setup** The ultrasonic electrostatic speaker (UES) (see Fig. 3(a)), placed at the center of a sensing module, broadcasts the ultrasonic chirp in the frequency range of 20-120 kHz with the maximum power of 78 dB SPL at 1 m. The UES is driven by a class AB speaker driver with a maximum power of 10 W. Four ultrasound condenser microphones (UCMs) are placed right, left, up, and down of the UES with the separation of 6 cm. The UCMs have a broad and flat frequency response in 20-150 kHz with the attenuation less than -6 dB. The recorder amplifies the received signals from the UCMs with the maximum gain of 40 dB and digitizes at a sampling rate of 750 kSample/s.

**Broadcasting Signal** Bats use a hyperbolic frequency-modulated (HFM) chirp containing multi harmonics, which has the effect of pulse compression increasing the spatial resolution as well as the receiver sensitivity assuring robust performance in environments with heavy reverberation [30–35]. Compared to the linear FM chirp, the HFM chirp is less sensitive to the frequency shifts caused by the movement of subjects because of its Doppler tolerance [36]. The waveform of the HFM chirp

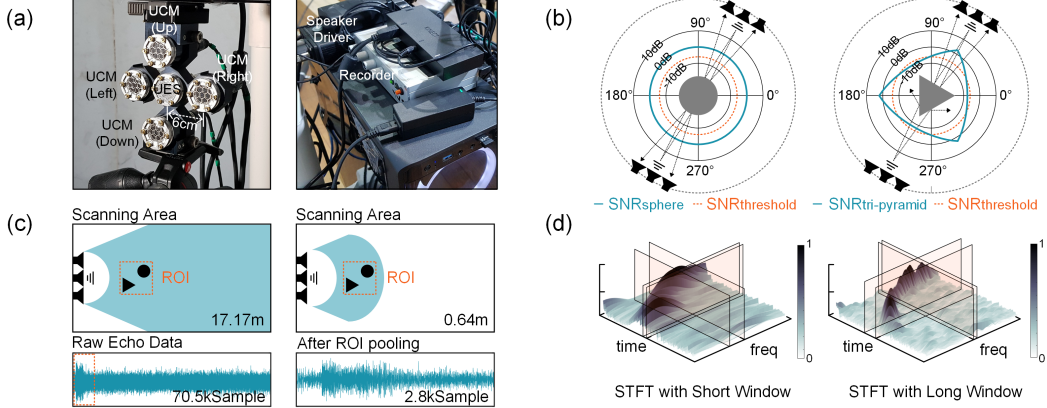

Figure 3: (a) Custom-made bat-inspired imaging (Bat-I) sensor for ultrasound data acquisition. (b) Polar plot of signal-to-noise ratio (SNR) of a sphere and a triangular pyramid. (c) The region-of-interest (ROI) pooling of the raw echo data. (d) Conversion of the echo signal into two spectrograms.

$x_{HFM}$ with pulse duration $T_{HFM}$, chosen as the broadcasting signal format in our Bat-I sensor, can be expressed as

$$x_{HFM}(t) = A(t) \cdot \sin\left[\frac{2\pi}{\xi}\ln(1 + \xi f_1 t)\right], \quad 0 \le t \le T_{HFM} \tag{4}$$

where $\xi = (f_1 - f_N)/f_1 f_N T_{HFM}$ and $f_1$ and $f_N$ are the first and the last carrier frequency, respectively, and $A(t)$ denotes a rectangular function given by $A(t) = a \cdot rect[(t - T_{HFM}/2)/(T_{HFM})]$[37]. The selected HFM parameters are $a = 0.3$, $T_{HFM} = 6$ ms, $f_1 = 120$ kHz, and $f_N = 20$ kHz.

**Objects and Data Acquisition** We have chosen 16.2 k geometric object configuration (such as cube, cone, sphere, and so on) as shown in Fig. 1(b), and created the objects using the building blocks and a 3D printer. The geometric objects are randomly placed in a $64^3$ cm$^3$ space with a distance of 1.48 m from the Bat-I sensor. Each target object is measured five times to desensitize the network to the ambient noise (e.g. noise from electronic equipment, footsteps, voice and so forth). We eventually acquired 81 k measured echo data.

**Data Processing** (1) *Thresholding* - An object reflects a limited portion of ultrasound energy back to the UCM. The backscattered power $\int_0^{T_s} |x_{r,i}(t)|^2 dt$ with scan duration $T_s$ received by the $i$-th UCM of a RADAR/SONAR system is

$$\int_0^{T_s} |x_{r,i}(t)|^2 dt = \frac{\int_0^{T_{HFM}} |x_{HFM}(t)|^2 dt \cdot G_t A_r \sigma e^{-2\alpha R_i}}{(4\pi)^2 R_i^4}, \quad i = 1, 2, ..., 4 \tag{5}$$

where $\int_0^{T_{HFM}} |x_{HFM}(t)|^2 dt$ and $G_t$ are the power of the transmitted HFM chirp and the gain of the UES, respectively [38, 39]. $A_r$ is the effective area of the UCM, $\sigma$ is the sonar cross section (SCS), $\alpha$ is the atmospheric attenuation constant, and $R_i$ is the distance from the UES/$i$-th UCM to the object. The SCS depends on the object's geometric shape, and orientation of the ultrasound source. In case an object has a small SCS (e.g. the most of the reflective surfaces of an object cause specular reflection), the SNR of the received signal drops below the minimum detectable SNR threshold (see Fig. 3(b)). As such, we have constructed training datasets with only reliable data that meet the threshold criteria such as

$$\mathbb{X}_{thr} = \{x_{r,i}^k(t) \mid \Delta_{dB}[x_{r,\forall}^k(t)] > -6 \ dB\}, \quad 0 \le t \le T_s \quad i = 1, 2, ..., 4 \tag{6}$$

where object-to-sphere power ratio (OSPR) of the $k$-th measured data of the $i$-th UCM $\Delta_{dB}[x_{r,i}^k(t)] = 10\log[\int_0^{T_s} |x_{r,i}^k(t)|^2 dt / \int_0^{T_s} |x_{sph}(t)|^2 dt]$. $\int_0^{T_s} |x_{sph}(t)|^2 dt$ is the backscattered power of an isotropic sphere (radius = 9 cm).

(2) *Pooling* - 70.5 k-Sample data recorded by the Bat-I sensor covers a scan depth of 17.17 m (the speed of sound $c = 343.42$ m/s) as described in Fig. 3(c). Processing raw data requires large computational resources. In order to reduce the input dimension, preliminary information reflecting

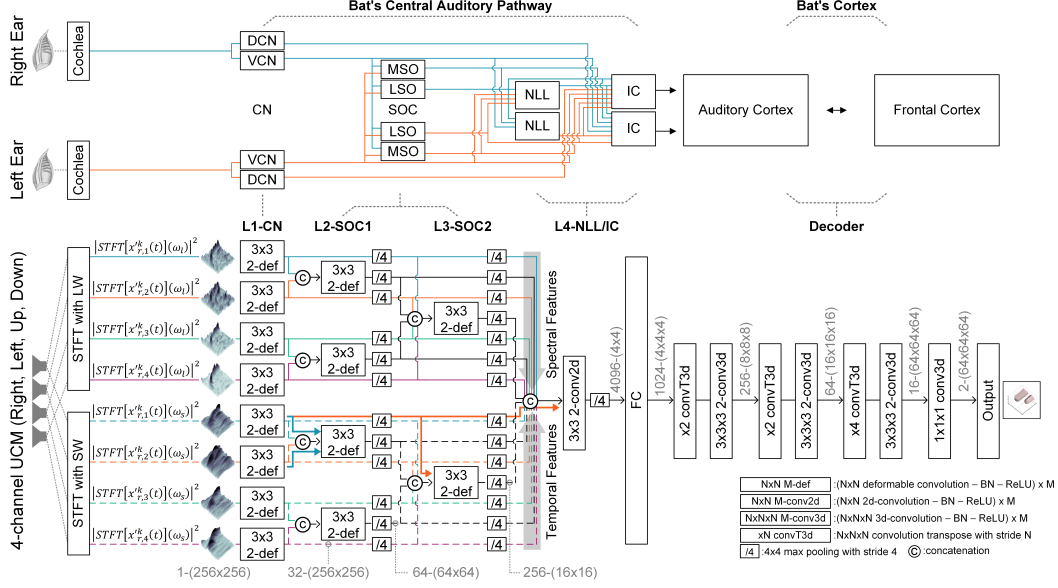

Figure 4: Simplified diagram of the anatomical connections of a bat's auditory system and architecture of the proposed Bat-G network (BN: batch normalization [40], ReLU: rectified linear unit [41]).

the fact that an object is placed at a distance of 1.48 m ± 32 cm is considered and only 2.8 k-Sample data covering the region-of-interest (ROI) are used. This process reduces the input dimension by 98 %. The ROI pooling can be expressed as

$$\mathbb{X}_{roi} = \{\acute{x}^k_{r,i}(t) = x^k_{r,i}(t + T_1) \mid x^k_{r,i} \in \mathbb{X}_{thr}\},$$
$$0 \le t \le T_2 - T_1 + \tau_{ir} \quad i = 1, 2, ..., 4 \quad (7)$$

where $T_1$ and $T_2$ are the start and the end time of the data covering ROI. $\tau_{ir}$ is the length of the intrinsic transient response of the UES/UCM.

(3) *Spectrogram* – Bat-G net includes two pathways primarily processing temporal or spectral cues similar to a bat's central auditory pathway (see section 5). In order to feed the appropriate signal to each path, the recorded signal is converted into two high-resolution spectrograms (see Fig. 3(d)) produced by the short-time Fourier transform (STFT) with a short/long hamming window $\omega_s/\omega_l$ (33-$\mu$s/133-$\mu$s window size with 22-$\mu$s/90-us overlap), namely

$$\mathbb{X}_{sp} = \{|STFT[\acute{x}^k_{r,i}(t)](\omega_s)|^2, |STFT[\acute{x}^k_{r,i}(t)](\omega_l)|^2 \mid \acute{x}^k_{r,i}(t) \in \mathbb{X}_{roi}\}.$$
$$0 \le t \le T_2 - T_1 + \tau_{ir} \quad i = 1, 2, ..., 4 \quad (8)$$

As the size of generated two spectrograms is different, they are resized to $256^2$. As a result, we have gathered 51.6 k data and each data is composed of eight spectrograms.

## 4.2  Labels (3D Ground-truth Model)

Each target object of the gathered data is modeled in 3D CAD and voxelized with the dimensions of $64^3$ (voxel size of $1^3$ cm$^3$). As the acoustic reflection coefficient at the interface between the air and the solid object material is close to one, the field of view (FoV) of the UCM is limited to the front view of the target objects. Therefore, shaded regions, from the back of the object to the end of the ROI, are padded by one.

## 5  Architecture of Proposed Bat-G Network

In this section, the architecture of the proposed Bat-G network that analyzes the 4-channel ultrasonic echoes and inversely reconstructs the 3D representation of the target objects is presented. The network consists of two components: (1) a neural encoder that emulates a bat's central auditory pathway

and (2) a 3D rendering decoder that is inspired by the expansive path of the U-net [42] without any concatenation from the contracting path.

## 5.1 Encoder

The 3D perception mechanism of the FM bats described in section 3 involves the neural interactions between the auditory pathway (through *the brainstem* and *the midbrain*) and *auditory cortex (AC)*. Fig. 4 depicts the simplified anatomical connections of a bat's auditory system (reconstructed from [43, 44]). The system consists of four main blocks[2]: (a) *VCN* where the cells (e.g. *the bushy* and *the octopus cells*) play an important role in extracting the timing information from the auditory nerve; and *DCN* where the principal neurons, including *the fusiform cells*, perform non-linear spectral analysis considering the location of the head and ears [46], (b) *SOC* (*MSO* and *LSO*) that calculates the interaural differences in time and intensity, contributing to the sound source localization, (c) *NLL* and *IC* where organized auditory information and the auditory nerve from peripheral brainstem nuclei converge, and (d) *AC* and *PFC* converting the integrated auditory features to a unified image. The architecture of the proposed Bat-G network emulates two features of a bat's auditory system.

(1) *Spectral/Temporal-Cue Dominant Path* - Some neurons are sensitive to the temporal- (time) or spectral- (frequency) domain information. These neurons form a nucleus, a cluster of neurons. Each nucleus intensively extracts domain-specific features depending on the nature of the neurons that make up the cluster. We constructed the front cluster of layers employing deformable convolution layer [47] which adjusts the receptive field according to the pattern of the temporal/spectral cues. In addition, the network pathway is divided into the two paths of dominantly processing either temporal or spectral cues of the input spectrogram.

(2) *Biomimetic Connections* - The nuclei directly or indirectly receive the monaural, binaural, ipsilateral, or contralateral signal from the lower auditory nuclei. In the aspect of network implementation, each ultrasonic echo spectrogram with a short/long window from four recording channels (right, left, up, and down) is monaurally processed at the corresponding *L1-CN* inspired by *CN* as shown in Fig. 4. The output feature-maps of *L1-CN*, *L2-SOC1*, or *L3-SOC2* are binaurally concatenated and then fed into one step deeper layers (highlighted in blue arrow). The feature maps are simultaneously transmitted to deeper layers (>2) via direct connection with successive stride-4 4 x 4 max pooling (highlighted in red arrow). *L4-NLL/IC* integrates entire products of each layer and then forwards the results (4 x 4 dimension vector with 4096 feature maps) to a 3D visualization decoder. The detailed structure (see Fig. 4) of each layer is as follows. The first three layers (*L1-CN*, *L2-SOC1*, and *L3-SOC2*) are implemented with two 3 x 3 deformable convolutions (dilation factor of 1, 3 x 3 offset, and "same" padding) followed by batch normalization (BN) [40] and rectified linear unit (ReLU) activation [41]. *L4-NLL/IC* consists of successive two conventional convolutions (3 x 3 kernels, and "same" padding with BN and ReLU).

## 5.2 Decoder

A 3D inverse rendering decoder projects the output data of the encoder in low dimensional manifold into the volumetric 3D image in $\mathbb{R}^{64 \times 64 \times 64}$ vector space (see Fig. 4). A fully-connected (FC) layer, applied to 4 x 4 pixel inputs encoded with 4096 feature maps, has 4096 hidden units. The output of the FC layer is reshaped into a 3D vector domain of $\mathbb{R}^{4 \times 4 \times 4}$ with 1024 feature maps. Then, the 3D vector passes through three 3D convolution transpose layers which are composed of one 3D convolution transpose (or deconvolution) layer (stride-2 2 x 2 x 2 or stride-4 4 x 4 x 4 kernels, and "same" padding with ReLU) and two 3D convolution layers (3 x 3 x 3 kernels, and "same" padding with BN and ReLU) . In order to convert a 16-feature vector into the desired representation, a 1 x 1 x 1 convolution layer is added to the final layer. The detailed structure of each layer is described in Fig. 4.

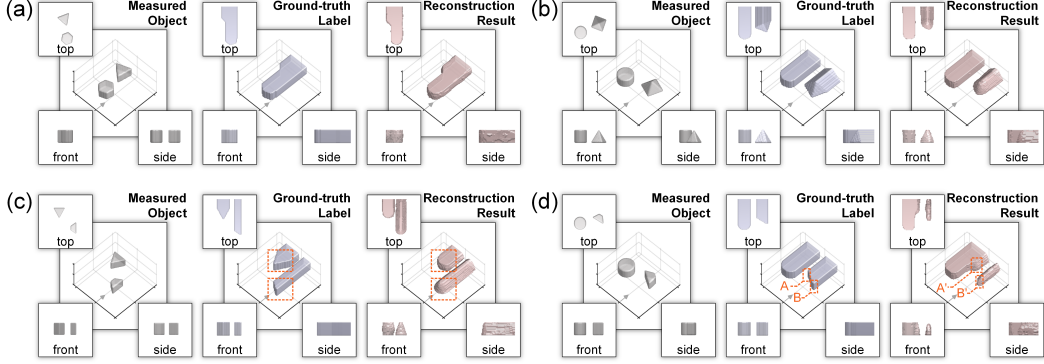

Figure 5: 3D reconstruction results of target objects when (a) the objects are composed of convex surfaces, (b) the objects have vertices, (c) the objects have an significantly small reflective area, and (d) the echo suffers the multiple diffusion reflections.

## 6 Training

The Bat-G network is trained employing a supervised learning algorithm. The network is repeatably fed with 49 k training data randomly selected from ECHO-4CH dataset (51.6 k data). The learning objective is minimizing the 3D reconstruction loss $L$ between the 3D network $f$ output $\hat{y} = f(\acute{x}_r) \in \mathbb{R}^{64 \times 64 \times 64}$, where $\acute{x}_r \in \mathbb{X}_{sp} \sim \mathbb{R}^{8 \times 256 \times 256}$ is the input spectrogram, and the corresponding ground truth label $y \in \mathbb{R}^{64 \times 64 \times 64}$. The loss function is implemented by employing $L_2$-regularization loss (regularization strength $\lambda = 10^{-6}$), and cross-entropy loss with softmax activation $S$, which can be expressed as

$$L(\hat{y}, y) = y \log[S(\hat{y})] + \lambda \sum_{i=1}^{m} \omega_i^2. \tag{9}$$

We adopted the Adam optimization algorithm [48] ($\beta_1$, $\beta_2$, and $\varepsilon$ are 0.9, 0.999, and $10^{-8}$, respectively) with an exponential decay (learning rate, decay rate, and decay steps are $10^{-4}$, 0.9, and 5 k, respectively) for better convergence. To reduce overfitting, dropout with the probability of retention of 0.5 [49] is applied to the network during training. The network is iteratively trained with 500 k steps on a GTX 1080 Ti GPU and a Threadripper 1900X CPU.

## 7 Experimental Results

We first present the qualitative assessment of the 3D rendering results of the Bat-G network. We then quantitatively evaluated 3D reconstruction performance based on precision, recall, and F1-score metrics. The Bat-G network is evaluated with 2.6 k test data of the ECHO-4CH dataset.

### 7.1 Qualitative Assessment

Fig. 5 shows the measured objects, the ground-truth labels, and the 3D reconstruction results of the Bat-G network presented in a 3D view and third angle projection. When a radiated ultrasonic chirp is reflected from convex surfaces of target objects, the 3D representation of the measured objects is uniformly reconstructed as shown in Fig. 5(a). It can be observed that the Bat-G network localizes the measured objects in a 3D-space and reconstructs the shapes of the objects by inferring based on test data. It is worth noting that the Bat-G network can reconstruct 3D shapes of the objects having vertices (Fig. 5(b)). Results presented in Fig. 5 clearly shows that the Bat-G net is sensitive to both azimuth and elevation cues. Examples yielding slightly unsatisfied outputs compared to those shown in Fig. 5(a)-(b) are presented in Fig. 5(c)-(d). From the reconstruction result in Fig. 5(c), it can be seen that the edge information of the measured objects are not fully-retrieved since the reflective area seen by the Bat-G sensor is significantly small. Fig. 5(d) shows that the ultrasound echo reflected from area $A$ is received through multiple diffusion reflection paths, while the reflected echo from area $B$ is measured primarily through the direct path. As a result, the Bat-G net erroneously represented the shape of $A$ because of the multiple diffusion reflections.

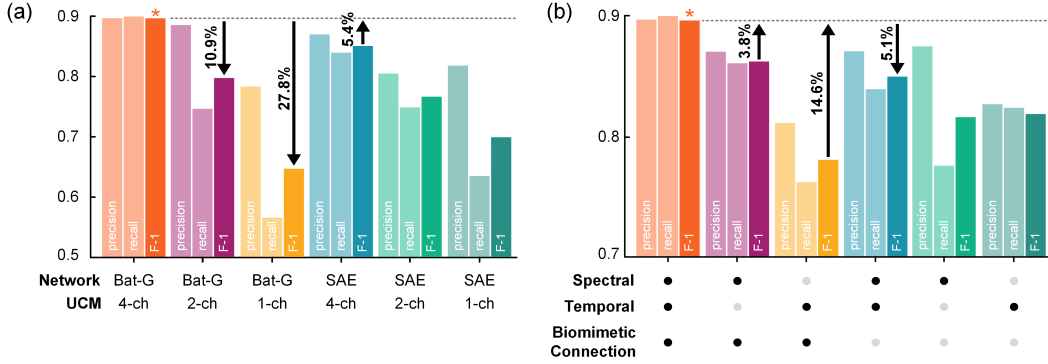

Figure 6: (a) Precision, recall and F1-score of the proposed Bat-G net and the stacked auto-encoder (SAE) employing the 4-, 2-, or 1-channel UCM input data. (b) Performance of Bat-G network with/without spectral/temporal-cue dominant path and/or the biomimetic connections.

## 7.2 Quantitative Assessment

As the volumetric 3D ground truth data is unbalanced (90 % of labels is label 0), the accuracy is always estimated to be higher than 90 % even though the network infers all outputs as label 0. Therefore, we quantitatively assessed the performance based on the precision, recall, and F1-score metric. The current state-of-the-art image reconstruction method using a neural network [50] demonstrates that the architecture composed of a conventional stacked auto-encoder (SAE) and FC layers can effectively learn forward reconstruction method composed of two manifold transformations: (a) diffeomorphism between sensory input and latent low dimensional space and (b) manifold mapping from latent space to the output image. Therefore, such SAE (with an FC layer) structure is employed as the baseline, while maintaining the number of parameters and layers equal to that of the Bat-G network for a fair comparison. The Bat-G net (4-channel UCM) achieves (see Fig. 6(a)) 0.896 in precision, 0.899 in recall, and 0.895 in F1-score which are 3.0 %, 7.1 %, and 5.4 % increase against the SAE (4-channel UCM), respectively. Besides, the contribution of the number of UCMs is assessed. As the number of UCMs decreases, the performance of the Bat-G net deteriorated (10.9 % and 27.8 % drops in F1-score when the number of UCMs reduces into two and one, respectively). This suggests that employing the 4-channel UCM data as the input is essential for the Bat-G net to reconstruct a 3D image that is sensitive to both azimuth and elevation cues. We also presented the ablation studies to validate efficacy of the spectral/temporal-cue dominant path and the biomimetic connection emulating a bat's auditory pathways. Employing both spectral and temporal pathways demonstrated the best performance, which means that the two pathways are complementary to each other (3.8 % or 14.6 % increases in F1-score against using only the spectral or the temporal pathway, respectively). When the biomimetic connection was removed, the performance degradation of 5.1 % was observed. The result shows that the nested biomimetic connection in the Bat-G net contributes significantly to extracting essential features required for 3D image reconstruction from ultrasonic echoes. More information on the network structures used for the comparison and the ablation studies can be found in the supplementary material.

## 8    Conclusion

In this study, a bat-inspired high-resolution 3D imaging system that can reconstruct the shape of target objects in 3D space using HFM ultrasonic echoes is presented. The proposed imaging system is composed of a Bat-G network and a Bat-I sensor that are equivalent to the central-auditory-pathway/auditory-cortex and the nose/ear of the bat, respectively. The Bat-G net was implemented using an encoder extracting temporal/spectral features from the hyperbolic chirped ultrasonic echoes, and a decoder reconstructing the 3D representation of a target object from the extracted features. The network is trained using a supervised learning algorithm with custom-made datasets (ECHO-4CH). Through a range of experiments, we have shown that the proposed network can effectively reconstruct the shapes of 3D objects. This work clearly demonstrates the implementation feasibility of a high-resolution ultrasound 3D imaging system used by live bats. It also marks a crucial step toward realizing an imaging sensor that can graphically visualize objects and their surroundings irrespective of environmental conditions, unlike conventional electromagnetic wave-based imaging systems.

**Acknowledgments**

The authors would like to thank Gain Kim, Soon-Won Kwon, and Sejun Jeon for their thoughtful comments on the manuscript. We thank all anonymous reviewers for their constructive feedback.

## Footnotes

[2]Note that acronyms of bat's nerve nuclei are listed in this footnote for the readability. *CN: the cochlear nucleus* (*VCN: the ventral CN* and *DCN: the dorsal CN*), *SOC: the superior olivary nuclei*, *MSO: medial superior olive*, *LSO: lateral superior olive*, *NLL: the nucleus of the lateral lemniscus*, *IC: the inferior colliculus*, *PFC: the prefrontal cortex*. [45]

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
