[Supplementary Material · NIPS2019_Supplementary.pdf]

# Supplementary - Bat-G net: Bat-inspired High-Resolution 3D Image Reconstruction using Ultrasonic Echoes

**Gunpil Hwang, Seohyeon Kim, and Hyeon-Min Bae**
School of Electrical Engineering
Korea Advanced Institute of Science and Technology
Daejeon, South Korea
{gphwang, dddokman, hmbae}@kaist.ac.kr

## A.  Additional Related Works

### A.1  Conventional Ultrasound Sensor

Ultrasonic sensors, called SONAR, have been used in the detection of submarines or shoals of fish, and used for the autonomous mobile robots to map and navigate in unknown and unstructured environments [1–5]. These ultrasonic sensors emit ultrasonic sounds with a single frequency and calculate the distance from the object by measuring the time-of-flight of echoes reflected from the object [6], and the accuracy of the measured distance can be enhanced by using frequency-modulated signals with a wideband frequency range [7–10]. It is possible to determine the position of an object in 3D space as well as distance by measuring the time difference between transmitter and receiver of several pairs, which is called 3D localization [11–15]. There are also sensors that apply beamforming to 3D localization by scanning the energy of reflected echoes in 3D space [16–20]. Efforts have been made to identify and classify the shape of the target object using ultrasound as well as the distance and position of the object. There have been approaches using the digitized envelope of amplitude of echoes [21], or High Range Resolution Profiles (HRRPs) [22], or the angle difference between the transducer and the object [12, 23], or the range-orientation and the amplitude-orientation relationships [24, 25] or ultrasound-tomography [26, 27]. However, the existing studies only classified the geometry elements such as planes, corners, and edges [12, 14, 23–25] and recognized the roughness of scattering surfaces such as flooring, carpets and asphalt [10, 24]. In contrast, our approach is able to locate the target and reconstruct the shape of geometric objects in 3D space beyond just recognizing and classifying the target object.

### A.2  Neural Network Approach

Various researches have been carried out using neural network to perform cross-domain conversion such as sensory-to-image conversion reconstruction. Such studies include the reconstruction of MRI image from the measurement of the sensor [28], the generation of an image of a player from the sound of a recorded instrument [29] and the tracking a person's pose with sensory data obtained with an RF sensor [30]. In addition, there have been attempts to apply neural networks to the echolocation or sonar, which includes the recognition of the acoustic image of 3D objects from the receiver arrays [31], the perception of a cube and a tetrahedron from the features of the spectrogram of the echoes [32], and the classification of the faces of the humans [33]. In this paper, We tried to convert the sensory data obtained from the ultrasonic sensor to the visual domain employing the neural network (referred to as Bat-G net) for the first time. The Bat-G network accomplished the 3D localization and the reconstruction of the shape of the target object.

## B.  Network Structures for Comparison and Ablation Studies

(a)

**Bat-G net (UCM 4-ch)**

(b)

**Bat-G net (UCM 2-ch)**

(c)

**Bat-G net (UCM 1-ch)**

Figure S1: Architectures of the Bat-G network employing (a) 4-channel UCM input data, (b) 2-channel UCM input data, and (c) 1-channel UCM input data

Figure S2: Architectures of the stacked auto-encoder (SAE) employing (a) 4-channel UCM input data, (b) 2-channel UCM input data, and (c) 1-channel UCM input data

Spectral +Temporal / Biomimetic

Spectral / Biomimetic

Temporal / Biomimetic

Figure S3: Architectures of the Bat-G network with (a) the spectral/temporal-cue dominant pathways, (b) the spectral-cue dominant pathway, and (c) the temporal-cue dominant pathway

Figure S4: Architectures of biomimetic-connection-ablated Bat-G network with (a) the spectral/temporal-cue dominant pathways, (b) the spectral-cue dominant pathway, and (c) the temporal-cue dominant pathway