[Reviews · NeurIPS 2019]

Reviewer 1



The paper appears to be rather original, in terms of task and architecture, although conventional in terms of machine learning theory. The proposed architecture appears to work reasonably well, and a few contrasting conditions are explored with varying performance. The main conditions are the so-called Bat-G network which features a late integration of spatial and temporal features, versus a so-called SAE network which performs earlier integration, and works less well. The effect of temporal versus spatial features is also compared. However the system contains a great many design decisions that are not explicitly tested. No conventional sonar/ultrasound methods are compared against, so it is difficult to determine how this work compares to prior methods of d al significance of the scores is rather unclear. For example, if the voxels at the visible surface were all correct, it seems the system could still achieve a bad score by misclassifying other voxels that are not visible. The writing is at times a bit ungrammatical (e.g., figure 5 caption "the objects have vertexes" instead of vertices). A glaring mistake is having figure 1 appear before the abstract. It is not possible to fully understand the architecture of the SAE network without seeing the supplementary material.

Reviewer 2



The proposed approach emits ultrasonic pulses and records the reflected echoes with multiple microphones. The audio input from the microphones is converted to spectrograms by using short-time fourier transform. The spectrograms are given as input to an encoder-decoder neural network architecture which is inspired by bat's auditory system and which is trained to output a volumetric voxel presentation of the scanned objects. The paper presents also a dataset where different objects are recorded with a custom-made echo scanner which can be rotated around the objects. The shape (and pose) of the scanned objects is known during data acquisition and can be used as a ground truth for training the network. I think that this paper addresses an interesting problem area which is apparently not much covered in machine learning previously. The proposed approach seems novel and contains lots of innovative design choices from experimental measurement setup to computational signal processing architecture. The obtained results look good. Since I am not an expert in the field of ultrasound or bats, I am a bit cautious to give strong recommendation.

Reviewer 3



Originality - I think the primary originality of the paper is limited to the engineering set-up specific to obtaining the dataset. It appears quite comprehensive and addresses a number of different geometric patterns that could prove to be a challenge for ultrasound-based image reconstruction. The biomimetic portion of the network also comes across as novel. Although, the specific impact of that portion of the network architecture is not clear. The rest of the paper applies standard supervised learning techniques to a labeled dataset and is not novel. Quality: This is a reasonably well written paper and attempts to solve a real-world problem. However, since the theme of the paper is a bat-inspired network, the paper fails to address the importance of the biomimetic part of the design. Figure 6 appears to perform ablation studies to judge the importance of the spectral and temporal and spectral cues. The authors claim that the comparison with SAE provides justification for the use of the biomimetic path. However, it would have been a more compelling argument if the performance of Bat-G was reported with the biomimetic connection removed. Clarity: the paper is well motivated and the experimental set-up is clearly described. The paper does a relatively poorer job of explaining the choice of baselines that were studied - e.g., why SAE, is it SotA on ultrasound image reconstruction? In the absence of such explanations, it is hard to judge the impact of the paper. Significance: I think the core significance of the paper comes from the large ultrasound dataset. The significance of the methods described in the paper is less clear. Post-rebuttal comments: I have increased my score based on the rebuttal.

[Author Response · NeurIPS 2019]

**We appreciate your time and valuable comments. We have carefully responded to all the reviewers' comments.**

**<Reviewer 2>** Q1. The impact on evaluation score when misclassifying invisible voxels - A1. As described in section 4.2, shaded regions (voxels that are not visible) are padded by one (as shown in Fig. AR1). Therefore, the invisible voxels do not affect the performance of the Bat-G network.

Q2. Grammatical mistake (e.g. vertexes) and Fig.1 appear before the abstract - A2. We will correct "vertexes" to "vertices" and move Fig. 1 to the second page. In addition, we will check other grammatical errors and fix them.

Measured Object $\oplus$ Shadow (Invisible) $=$ Ground-truth Label

Figure AR1

Q3. Hard to fully understand the SAE network without seeing the supplementary material - A4. In the final manuscript, we will gladly include the specific details of the SAE (from the supplementary material) and the reason why we have chosen the SAE as a baseline (from Q1 – Reviewer 4) in the main manuscript.

**<Reviewer 3>** Q1. Comparison between 3D US imaging in medicine and the proposed system - A1. The fundamental differences in principle and implementation are depicted in Fig. AR2 due to the page limit.

Figure AR2

Q2. Compressing Sections 1-3 and adding more details to Sections 5-7 - A2. We will reorganize the contents of each section to add more details on the design strategies of the Bat-G net and analysis/discussion of the experimental results.

**<Reviewer 4>** Q1. Why SAE was chosen as the baseline - A1. The current state-of-the-art image reconstruction method using a neural network [Bo Zhu et al., 2018] demonstrates that the architecture composed of SAE and fully-connected (FC) layers can effectively learn forward reconstruction method composed of two manifold transformations: (a) diffeomorphism between sensory input and latent space in low dimension and (b) manifold mapping from latent space to the output image. Therefore, such SAE (with a FC layer) structure is selected as the baseline, while maintaining the number of parameters and layers equal to that of Bat-G network for a fair comparison. In the final manuscript, we will add the description of the reason why we have chosen the SAE structure as the baseline.

Q2. It would have been a more compelling argument if the performance of Bat-G was reported with the biomimetic connection removed - A2. Biomimetic connection emulating a bat's auditory pathways includes monaural/binaural path and direct connection to deeper layers (>2). When the biomimetic connections are removed from the Bat-G network as shown in Fig. AR3, the ablated output eventually becomes identical to the SAE. As a result, comparing the reconstruction performance of the Bat-G net with that of SAE is equivalent to validating the efficacy of biomimetic connections of the Bat-G net.

Figure AR3

[Meta-Review · NeurIPS 2019]

The reviews are moderately positive and the AC recommends accepting this paper as a poster. The authors are requested to address the reviewers' concerns and integrate material from the rebuttal into the revision.